# Changing perception and improving knowledge of leprosy: An intervention study in Uttar Pradesh, India

Anna T. van 't Noordende[1,2]*, Suchitra Lisam[3], Vivek Singh[3], Atif Sadiq[3], Ashok Agarwal[3], Duane C. Hinders[1], Jan Hendrik Richardus[2], Wim H. van Brakel[1], Ida J. Korfage[2]

**1** NLR, Amsterdam, The Netherlands, **2** Department of Public Health, Erasmus MC, University Medical Center Rotterdam, Rotterdam, The Netherlands, **3** NLR India, New Delhi, India

* a.vt.noordende@nlrinternational.org

## Abstract

### Introduction

Since ancient times leprosy has had a negative perception, resulting in stigmatization. To improve the lives of persons affected by leprosy, these negative perceptions need to change. The aim of this study is to evaluate interventions to change perceptions and improve knowledge of leprosy.

### Methodology/Principal findings

We conducted a pre-post intervention study in Fatehpur and Chandauli districts, Uttar Pradesh, India. Based on six steps of quality intervention development (6SQuID) two interventions were designed: (1) posters that provided information about leprosy and challenged misconceptions, and (2) meetings with persons affected by leprosy, community members and influential people in the community. The effect of the interventions was evaluated in a mixed-methods design; in-depth interviews, focus group discussions, and questionnaires containing a knowledge measure (KAP), two perception measures (EMIC-CSS, SDS) and an intervention evaluation tool. 1067 participants were included in Survey 1 and 843 in Survey 2. The interventions were effective in increasing knowledge of all participant groups, and in changing community and personal attitudes of close contacts and community members (changes of 19%, 24% and 13% on the maximum KAP, EMIC-CSS and SDS scores respectively, p<0.05). In Survey 1, 13% of participants had adequate knowledge of leprosy versus 53% in Survey 2. Responses showed stigmatizing community attitudes in 86% (Survey 1) and 61% (Survey 2) of participants and negative personal attitudes in 37% (Survey 1) and 19% (Survey 2). The number of posters seen was associated with KAP, EMIC-CSS and SDS scores in Survey 2 (p<0.001). In addition, during eight post-intervention focus group discussions and 48 interviews many participants indicated that the perception of leprosy in the community had changed.

---

**Data Availability Statement:** The Excel database are available from the infolep website through the following link: https://www.infontd.org/sites/default/files/2021-07/210726%20Database%

20changing%20perception%20and%20improving
%20knowledge%20of%20leprosy.txt.

**Funding:** This study is part of a larger research project, the Post-Exposure Prophylaxis (PEP++) project. The PEP++ project is funded by the Dream Fund of the Dutch Postcode Lottery. The funders had no role in study design, data collection and analysis, decision to publish, or preparation of the manuscript.

**Competing interests:** The authors have declared that no competing interests exist.

## Conclusions/Significance

Contextualized posters and community meetings were effective in changing the perception of leprosy and in increasing leprosy-related knowledge. We recommend studying the long-term effect of the interventions, also on behavior.

### Author summary

To improve the lives of persons with leprosy, perceptions about leprosy need to change. The aim of this study is to describe the development and evaluation of interventions (posters and community meetings) to change the perception and improve knowledge of leprosy in Fatehpur and Chandauli districts, India. To measure the effect of the interventions we administered questionnaires before and after the interventions and we conducted 48 interviews and eight group interviews afterwards. In total 1067 participants were included in the first survey and 843 in the second. We found that the interventions resulted in more knowledge of leprosy and in changed community and personal attitudes towards (persons affected by) leprosy. The percentage of participants with adequate knowledge of leprosy was higher in the second survey and the percentage of participants with negative community and personal attitudes was lower. In the interviews, many participants indicated that there had been a change in perception in the community. The more posters participants had seen, the better their knowledge of leprosy and the more positive their attitudes. Findings from this study suggest that contextualized posters and community meetings can be effective in changing the perception of leprosy and increasing leprosy-related knowledge.

## Introduction

Perception is a broad concept, that refers to how an individual or group "sees" an object, person, event or institution [1–3]. Perception encompasses how an individual or group "sees" others (social perception), but also a person's interpretation and understanding of a disease and its potential consequences (disease perception) [1–3]. Perception comprises knowledge, beliefs, attitudes and emotions that are in turn influenced by personal factors (e.g. personality, experience) and environmental factors (e.g. culture, religion) [1, 2]. These concepts are interrelated. (Negative) perception is related to stigma. However, where perception is solely cognitive, stigma includes both cognitive (e.g. knowledge, attitudes, labelling) and behavioral (e.g. discrimination, rejection, withdrawal) elements [4, 5]. Perception is an important driver of stigma [6].

Leprosy is an infectious disease that has had a negative perception, resulting in stigmatization, since ancient times [7]. The main causes of leprosy-related stigma are the external manifestations of the disease (such as impairments of eyes, hands and feet), religious and cultural beliefs, fear, and a lack of knowledge [7, 8]. Almost all areas of a person's life can be affected by stigma, such as employment and education opportunities, social interaction, marriage (prospects), housing and access to care [9]. These negative consequences and the fear of being stigmatized can cause chronic stress, which may negatively impact mental wellbeing and physical health [9]. In the case of health-related stigma, the fear of being stigmatized may also cause people to delay or avoid seeking treatment or care [9]. To improve the lives of persons affected

 

by leprosy and to improve leprosy services, negative perceptions about leprosy need to be addressed.

There are several strategies and interventions to change the perception of leprosy. Many of these are similar or the same as interventions for stigma reduction. Interventions that aim to reduce stigma often address the causes of stigma, such as beliefs and attitudes that lead to labelling, stereotyping and discrimination [10]. Interventions that have reduced leprosy-related stigma include 'contact events' in which contact between persons affected by leprosy and community members is enhanced, socioeconomic rehabilitation, peer counselling, social marketing campaigns, community engagement interventions and mass media campaigns [11–16]. Crucial to changing perceptions is understanding the local context, and understanding and addressing the drivers of these perceptions [10, 17, 18]. Interventions should fit the audience [19]. They are more likely to be successful if culture-specific and contextualized (adapted to the local context) [20, 21], addressing the main causes of leprosy: specific knowledge gaps, beliefs and fears [22].

The present study is part of a project on leprosy prevention in India, Indonesia and Brazil, the PEP++ project (https://www.trialregister.nl/trial/7022). The aim of the present study is to evaluate interventions to change perception and improve knowledge of leprosy.

## Definitions

Perception comprises knowledge, beliefs, attitudes and emotions. Beliefs link an object (such as a person, group of people, disease, institution or behaviour) to an attribute. For example the belief "leprosy is dangerous", links "leprosy" (object) to "dangerous" (attribute) [16]. Knowledge refers to theoretical or practical understanding of a subject (facts, skills or objects). Truth and belief are a prerequisite for possessing knowledge: one has a belief in something, and that belief must be true (based on observable and measurable evidence). For example, if you know that leprosy is an infectious disease, then you must believe this, and your belief must be true [17]. An attitude refers to a person's feelings toward and evaluation of an aspect of the person's world, for example an object, person, event, or towards performing specific behaviours [16, 18]. It refers to "a person's location on a dimension of affect or evaluation" and falls on a continuum from very favourable to very unfavourable [16]. Emotions are inner states such as anger, joy, fear or love. Emotions can be consciously experienced, but can also be repressed, inhibited or unconscious [21].

## Methods

### Ethics statement

Ethical approval for this study was obtained from the Vardhman Mahavir Institutional Ethics Committee as part of a larger research project: the PEP++ project. Written informed consent for participation was obtained from each participant prior to data collection.

### Study setting

The study was conducted in two districts in Uttar Pradesh, northern India: Chandauli (population 1.95 million, 1548 villages) and Fatehpur (population 2.63 million, 1476 villages). These districts have a relatively high number of new leprosy patients annually with a new case detection rate of 5.9 per 100,000 population in both Chandauli and Fatehpur, in March 2019 (District Leprosy Office).

## Study design

We applied a pre/post intervention study design. The effect of the interventions was evaluated using mixed methods.

## Eligibility criteria

We included four groups as participants in the study: (1) persons diagnosed with leprosy at any time ("persons affected by leprosy"); (2) close contacts of persons affected by leprosy, these comprised household contacts, family members, neighbours and other social contacts; (3) community members; and (4) health care workers. Only individuals 16 years or older were included. Close contacts, community members and health care workers were excluded if they had ever been diagnosed with leprosy.

## Interventions

The PEP++ project includes interventions that aim to change the perception of leprosy, improve knowledge of leprosy and reduce stigma, and to increase the community acceptance of preventive (chemoprophylactic) treatment. These interventions provide contextualized (adapted to the local context) information, education and communication (IEC) and are implemented before the implementation of the actual chemoprophylaxis. In doing so, we aim to increase acceptance and adherence to preventive chemoprophylactic treatment. The interventions were designed based on the six steps of quality intervention development (6SQuID) [23] using a community engagement method. 6SQuID is a pragmatic guide, based on existing frameworks for the development of interventions with a wider public health focus. The main input for the selection of the interventions (content and modes of delivery) came from: a) 'leprosy perception study' (Survey 1) of people's knowledge and perceptions of leprosy and persons affected by leprosy [8, 22, 24], b) a 'communication needs assessment', and c) a workshop with input from persons affected by leprosy and other key stakeholders. A detailed description of the selection and development, including the pilot tests, of the interventions can be found as supporting information file (S1 Text).

Two interventions are assessed in this paper: (1) posters and (2) community meetings. Posters were available in three sizes (46x58 cm, 44x14 cm and 28x23 cm) and covered the following themes: symptoms, mode of transmission, cause, curability, (free) treatment, prevention of leprosy, and inclusion of persons affected by leprosy in the community. The posters and an English translation can be found as supporting information file (S2 Text). The villages in which the posters were put up were selected based on the number of leprosy patients registered at the health center since April 2014. Only villages in which at least two patients were registered were selected. The posters were placed at several locations in the villages (e.g., at the village leader's house, shops, the health facility, crossroads, ATMs, temples and the marketplace) and near sites of public transport (e.g., in buses and auto rickshaws, and at bus stops and railway stations).

The community meetings were held in villages selected from the list of 606 villages in which posters were put up. Villages in which the Pradhan (village leader) was available on scheduled meeting days and where the prior relationship with the Pradhan was good, were selected. Community members were invited to attend the meeting through the Pradhan and by door-to-door visits from community health workers (ASHA's). The meetings itself consisted of a short presentation about leprosy and the PEP++ project, followed by questions-and-answers and a discussion. In some meetings, two short videos about leprosy were also presented (due to technical issues this was not possible in all meetings). A health worker was present during the meetings. Meetings with key influential people were held at district or block level, while

meetings with community members and persons affected by leprosy were held in the communities. Participants were also offered a leaflet with more information about leprosy (facts, myths and misconceptions) and the PEP++ project at the meetings. An overview of the reach of the two interventions can be found in Table 1. We report on perception and knowledge of leprosy before and after the interventions.

## Outcomes

Four outcome measures were used to assess perception: (1) a knowledge, attitudes and practices (KAP) measure; (2) the Explanatory Model Interview Catalogue Community Stigma Scale (EMIC-CSS); (3) the Social Distance Scale (SDS); and (4) an intervention evaluation tool to assess exposure to the posters. In addition, in-depth interviews and focus group discussions were conducted.

The KAP measure was used to assess the knowledge, attitudes and practices of participants regarding leprosy. On some of the questions, multiple answers were possible. A maximum score of eight could be obtained on the KAP if all correct answers are provided, even if incorrect answers were present. We defined 'poor knowledge' as a score of two or less out of eight, 'moderate knowledge' as a score between two and six and 'adequate knowledge' as a score of six or more on the KAP. These cut-offs were chosen arbitrarily, as no external criterion was available. The KAP measure has been used in several leprosy studies in Nepal, India, Indonesia and Brazil between 2012 and 2018 [8, 25–27].

The EMIC-CSS was used to measure perceived community attitudes and behavior towards persons affected by leprosy. A total maximum score of 30 can be obtained, ranging from zero (no negative attitudes) to 30 (most negative attitudes). The EMIC-CSS has been validated among community members of persons affected by leprosy in India [28]. We operationalized stigmatizing community attitudes towards leprosy as a sum score of 8 or higher on the EMIC-CSS, using the cut-off point of 8 that was proposed by Sermrittirong and colleagues [29].

The SDS was used to assess the social distance the participant wants to keep towards persons affected by leprosy. This measure was used as a proxy for personal attitudes and fears of

**Table 1. An overview of the interventions, their target groups, periods of dissemination.**

| Intervention | Target group | Time period disseminated |
|---|---|---|
| **Posters**<br>16,070 large size posters (46x58 cm) and 8,384 smaller size posters (4,192 size 44x14 cm and 4,192 size 28x23 cm) were put up in 606 villages across the two districts. A total of six different formats (different images and key messages) were used. | • Persons affected by leprosy<br>• Close contacts<br>• Community members | October 2019—April 2021 (ongoing for full project duration) |
| **Community meetings**<br>271 meetings were held across the two districts, reaching a total of 12,933 people. A total of 9,421 leaflets were disseminated at the meetings. Separate meetings were held per target group.<br>Of the 271 meetings held:<br> • 128 meetings were held for key influential people in the community, reaching 2,840 people<br> • 98 meetings were held for community members, reaching 7,668 people<br> • 12 "Shiv charcha" (religious) meetings were held in Chandauli, reaching 1,429 people<br> • 33 meetings were held for persons affected by leprosy in Fatehpur, reaching 996 persons affected by leprosy. | • Key influential people in the community (teachers, informal practitioners, heads of the village, religious leaders and media personnel)<br>• Community members<br>• Persons affected by leprosy | December 2019 -February 2020 |

the respondent. The SDS has 7 questions on which a maximum score of 21 can be obtained, ranging from zero (no negative attitudes) to 21 (most negative attitudes). The SDS has been translated to Hindi and was partially validated among community members of persons affected by leprosy in Uttar Pradesh, India [25]. We chose a cut-off for negative personal attitudes when participants answered at least 3 questions with 'probably not willing,' or at least one question with 'definitely not willing' and at least one question with 'probably not willing.'

The intervention evaluation tool consisted of questions about exposure to the posters. For example, participants were shown the posters and asked whether they had seen them recently, and participants were asked to identify correct messages about leprosy (read aloud while shown on the posters). The EMIC-CSS and SDS assess community stigma and were therefore not administered to persons affected by leprosy. The intervention evaluation tool was not administered to health workers, because they were not a target group for the posters and meetings.

Semi-structured in-depth interviews and focus group discussions were conducted for insight into specific local beliefs, myths and misconceptions of the participants towards leprosy and persons affected by leprosy. The interview guide was pilot tested before use [8].

## Participant timeline

Survey 1 was conducted between March 2017 and December 2018. The outcomes represent the pre-intervention (baseline) information. After finalization of the interventions, dissemination of posters started from October 2019 and onwards, and the community meetings were held between September 2019 and February 2020. The post-intervention or evaluation study (Survey 2) was conducted between March and June 2020. An overview of the study design can be found in Fig 1.

## Sample size

We aimed to include a random sample of at least 100 persons of each target group per district for the interview-administered questionnaires. This is based on an assumed prevalence of, for example, 'negative attitudes' of 50% at baseline and wanting to be able to detect a reduction of 20% or more (meaning that the prevalence of negative attitudes in the second survey is 30% or less). Using these parameters, a significance level of 0.05 and a power of 80%, 93 participants are needed before and 93 are needed after the intervention is implemented (calculated using Epi Info StatCalc for cross-sectional studies). To compensate for records that may not be usable or loss to follow up, we aimed to include at least 100 participants per target group. The data of community members for Survey 1 were collected in a separate study but using the

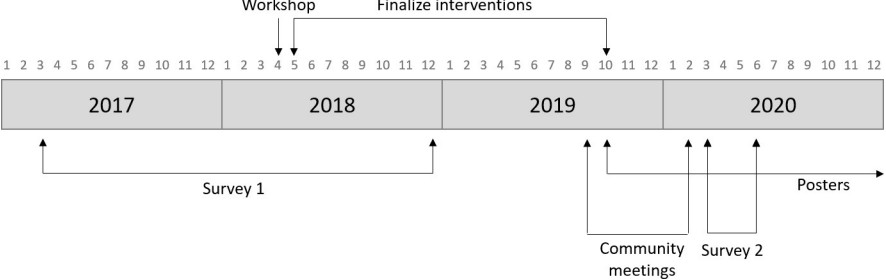

**Fig 1. Study design and timeline.** Survey 1 consisted of the KAP, EMIC-CSS, SDS, communication needs assessment, in-depth interviews, and focus group discussions. Survey 2 consisted of the KAP, EMIC-CSS, SDS, intervention evaluation tool, in-depth interviews, and focus group discussions.

same instruments, in the same area and timeframe. Thus, these data were included instead interviewing another sample of community members.

## Recruitment and sampling procedure

For pre- and post-intervention assessments, the persons affected by leprosy were selected by stratified systematic sampling with a random start from a list of leprosy patients registered at the primary health care center. Close contacts of leprosy patients and community members were selected by convenience sampling from among those living in the same village or neighborhood as the person affected by leprosy. Health care workers were selected based on convenience sampling from among those present and available at the primary health care centers. Half of the health care workers had received training about leprosy and had specific responsibilities for leprosy treatment services. Details about the selection procedure have been published previously [8, 22]. Different (randomly selected) participants were included in the first and second survey.

In addition, in each district we aimed to include six persons from each participant group in the in-depth interviews (IDI) and to conduct one focus group discussion (FGD) per participant group. These participants were a subset of those in the quantitative sample.

## Data collection

Data for the perception studies (Survey 1 and Survey 2) were collected before and after the posters were distributed and community meetings were held. In Survey 2, additional demographic information was collected from the participants about income and caste. In addition, Survey 2 data were collected in the areas in which interventions were conducted. Participants were interviewed by a trained research assistant at or near their homes, at primary health centers or the district offices of NLR India. Details of Survey 1 have been published previously [8, 22].

## Data management

All participants provided informed consent prior to data collection. The hard copy informed consent forms are stored in a locked archive in the field offices of NLR India. Questionnaire data were collected on paper and a Data Entry Officer subsequently entered the responses into a database created in Epi Info. The interviews were recorded on a voice recorder and transcribed in Microsoft Word. The audio files of the interviews were deleted after transcription and data analysis.

## Data analysis

Data analysis of the quantitative data were performed in SPSS. No records needed to be excluded from analysis. Simple descriptive methods were used to generate a demographic profile of the study sample. Differences between participants in the first and second survey were evaluated using an independent samples t-test for continuous variables (age) and Chi-square statistics for categorical variables. Corrected median differences and the statistical significance of changes ($p < 0.05$) in KAP, EMIC-CSS and SDS scores between Survey 1 (before any intervention) and Survey 2 (after the posters and community meetings) were calculated using quantile regression in which we corrected for age, sex, district, education, religion, participant type and data collection period (Survey 1 or Survey 2). Correlations between exposure to the posters and KAP, EMIC-CSS and SDS scores were calculated using Spearman's rank correlation. Because of the differences in participants in Survey 1 and 2, we could not use a

standardized method for effect size. We therefore used the corrected median difference in scale scores between Survey 1 and 2 as percentage of the maximum score that can be obtained on the KAP, EMIC-CSS and SDS to indicate the magnitude of the effect of the interventions.

In addition, we used stepwise multivariate regression with backward elimination to investigate the contribution of potential determinants (age, gender, participant type, marital status, education, occupation, knowing someone affected by leprosy, district and total number of posters seen) to the outcomes of interest (knowledge, stigma, social distance) for dependent variables that were normally distributed. We used bootstrapped stepwise multivariate regression with backward elimination for dependent variables that were not-normally distributed. Only variables that had a $p$-value of $\leq 0.2$ in univariate analysis were considered for entry into the multivariable regression model. Variables were eliminated from the multivariate model one-by-one until only statistically significant variables ($p<0.05$) remained.

The in-depth interviews and focus group discussions were audio recorded, transcribed verbatim and translated from Hindi to English. The data were analyzed using open, inductive coding and content analysis. Qualitative data analyses were performed in the software program MAXQDA. All records were anonymized before analysis.

## Results

### Socio-demographic information

In total 1067 participants were included in the first survey and 843 participants in the second survey; see Table 2 for an overview of the demographic information of the participants. Statistically significant differences (p<0.05) between participants in the first and second survey

**Table 2. Overview of the demographic characteristics of the participants included in Survey 1 (before any intervention, n = 1067) and Survey 2 (after the posters and community meetings, n = 842).**

| Variable | Persons affected by leprosy | | | Close contacts and community members | | | Health workers | | |
|---|---|---|---|---|---|---|---|---|---|
| | Survey 1 (n = 200) | Survey 2 (n = 201) | p-value[a] | Survey 1 (n = 767) | Survey 2 (n = 541) | p-value[a] | Survey 1 (n = 100) | Survey 2 (n = 101) | p-value[a] |
| Age, mean (SD) | 39.1 (15.7) | 41.9 (16.1) | 0.087 | 40.5 (16.1) | 36.7 (13.8) | <0.001 | 41.8 (11.1) | 40.8 (10.7) | 0.522 |
| Sex, *n* (%) | | | 0.385 | | | 0.021 | | | 0.001 |
| Female | 77 (38.5) | 69 (34.3) | | 297 (38.7) | 244 (45.1) | | 59 (59.0) | 35 (35.0) | |
| Male | 123 (61.5) | 132 (63.7) | | 470 (61.3) | 297 (54.9) | | 41 (41.0) | 65 (65.0) | |
| District, *n* (%) | | | 0.960 | | | <0.001 | | | 1.000 |
| From Fatehpur | 100 (50.0) | 101 (50.2) | | 296 (38.6) | 271 (50.1) | | 50 (50.0) | 50 (50.0) | |
| From Chandauli | 100 (50.0) | 100 (49.8) | | 471 (61.4) | 270 (49.9) | | 50 (50.0) | 50 (50.0) | |
| Education completed, *n* (%) | | | 0.020 | | | <0.001 | | | 0.317 |
| No (formal) | 72 (36.0) | 85 (42.3) | | 207 (27.0) | 184 (34.0) | | 0 (0.0) | 0 (0.0) | |
| Primary | 23 (11.5) | 40 (19.9) | | 104 (13.6) | 115 (21.3) | | 0 (0.0) | 1 (1.0) | |
| Secondary or higher | 105 (52.5) | 76 (37.8) | | 456 (59.5) | 242 (44.7) | | 100 (100.0) | 99 (99.0) | |
| Religion, *n* (%) | | | 0.841 | | | 0.001 | | | 1.000 |
| Hinduism | 184 (92.0) | 186 (92.5) | | 687 (89.6) | 512 (94.6) | | 99 (99.0) | 99 (99.0) | |
| Islam | 16 (8.0) | 15 (7.5) | | 72 (9.4) | 28 (5.2) | | 1 (1.0) | 1 (1.0) | |
| Other | 0 (0.0) | 0 (0.0) | | 8 (1.0) | 1 (0.2) | | 0 (0.0) | 0 (0.0) | |
| Marital status, *n* (%) | | | 0.159 | | | <0.001 | | | 0.623 |
| Currently married | 150 (75.0) | 162 (80.6) | | 313 (40.8) | 425 (78.6) | | 91 (91.0) | 89 (89.0) | |
| Never married | 38 (19.0) | 32 (15.9) | | 74 (9.6) | 97 (17.9) | | 9 (9.0) | 10 (10.0) | |
| Other[b] | 12 (6.0) | 7 (3.5) | | 9 (11.7) | 19 (3.5) | | 0 (0.0) | 1 (1.0) | |
| Missing | 0 (0.0) | 0 (0.0) | | 371 (48.4) | 0 (0.0) | | 0 (0.0) | 0 (0.0) | |

[a]The p-value is based on independent samples t-test for continuous variables (age) and X2 statistics for categorical variables.

[b] Marital status 'other' refers to participants who are separated, divorced or widowed.

were found for education level of persons affected by leprosy (participants in the second survey had in general had less education), for gender of health workers (more men were included in the second survey), and for all demographic variables of close contacts and community members.

In addition, after the intervention eight focus group discussions were conducted with 62 participants in total (one focus group with each target group in each district, n = 47 male and n = 17 female, average age 40 years, range 20–80 years) and 48 in-depth interviews (six per target group in each district, n = 25 male and n = 23 female, average age 37 years, range 19–58 years).

## Overall impact of the interventions: difference between Survey 1 and 2

In Survey 1 the percentage of participants with adequate knowledge on the KAP measure was 13% (n = 133; 13% of the persons affected by leprosy, 7% of the contacts and community members, 56% of the health workers). In Survey 2, 53% (n = 448) of the participants had adequate knowledge of leprosy (78% of the persons affected by leprosy, 38% of the contacts and community members, 87% of the health workers). An overview of the distribution of the KAP measure scores in Survey 1 and 2 of the contacts and community group can be found in Fig 2. In addition, in Survey 1, 86% of participants had stigmatizing attitudes towards leprosy on the EMIC-CSS (n = 747; 86% of the contacts and community members, 84% of the health workers). In Survey 2, this was 61% (n = 393; 65% of the contacts and community members, 43% of the health workers). In addition, 37% of participants had negative personal attitudes on the SDS in Survey 1 (n = 325; 41% of the contacts and community members, 14% of the health workers) and 19% in Survey 2 (n = 121; 22% of the contacts and community members, 2% of the health workers).

Tables 3 and 4 provide an overview of the differences in KAP, EMIC-CSS and SDS scores between Survey 1 and Survey 2 per district and per participant group. Compared to Survey 1, almost all KAP, EMIC-CSS and SDS scores improved. The scores that did not improve statistically significantly were the EMIC-CSS score for health workers in Fatehpur and the KAP and SDS scores for health workers in Chandauli (Tables 3 and 4). When looking at the corrected median differences for all participants groups (the 'whole dataset' rows in Tables 3 and 4), the corrected median difference was 1.5 for the KAP (a change of 19% of the maximum score of 8 that can be obtained on the scale), -7.3 for the EMIC-CSS (a change of 24% of the total score of

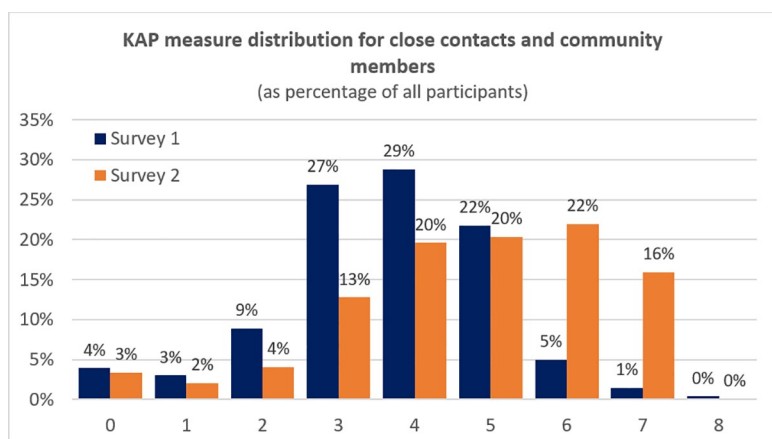

**Fig 2. Distribution of the KAP measure scores for close contacts and community members in Survey 1 and Survey 2.**

**Table 3. Corrected median differences in KAP (range 0–8) scores between Survey 1 (n = 1067) and Survey 2 (n = 842).**

| Dataset | KAP measure | | | | |
|---|---|---|---|---|---|
| | *Survey 1, median (Q1-Q3)* | *Survey 2, median (Q1-Q3)* | *Corrected median difference[a]* | *Change in score[b]* | *p-value[c]* |
| *Whole dataset (n = 1067~842)* | **4.0 (3.0–5.0)** | **6.0 (4.0–7.0)** | **1.5** | **18.8%** | **<0.001** |
| Persons affected by leprosy (n = 200~201) | 4.0 (3.0–5.0) | 7.0 (6.0–8.0) | 3.0 | 37.5% | <0.001 |
| Contacts and community (n = 767~541) | 4.0 (3.0–5.0) | 5.0 (4.0–6.0) | 1.0 | 12.5% | <0.001 |
| Health workers (n = 100~100) | 6.0 (5.0–7.0) | 7.0 (6.0–7.0) | 1.0 | 12.5% | <0.001 |
| *Chandauli district (n = 621~420)* | **4.0 (3.0–5.0)** | **6.0 (5.0–7.0)** | **2.0** | **25.0%** | **<0.001** |
| Persons affected by leprosy (n = 100~100) | 3.0 (3.0–4.0) | 6.0 (5.0–7.0) | 3.0 | 37.5% | <0.001 |
| Contacts and community (n = 471~270) | 4.0 (3.0–5.0) | 5.0 (4.0–6.0) | 2.0 | 25.0% | <0.001 |
| Health workers (n = 50~50) | 6.0 (5.0–7.0) | 7.0 (7.0–7.0) | 0.0 | 0.0% | NS |
| *Fatehpur district (n = 446~422)* | **4.0 (3.0–5.0)** | **5.0 (4.0–7.0)** | **1.3** | **16.3%** | **<0.001** |
| Persons affected by leprosy (n = 100~101) | 4.0 (3.0–5.0) | 7.0 (6.0–8.0) | 3.0 | 37.5% | <0.001 |
| Contacts and community (n = 296~271) | 4.0 (3.0–5.0) | 4.0 (3.0–6.0) | 1.0 | 12.5% | <0.001 |
| Health workers (n = 50~50) | 5.5 (4.0–6.3) | 7.0 (6.0–7.0) | 1.0 | 12.5% | 0.014 |

[a] We corrected (adjusted) for age, sex, district, education, religion, participant type and data collection period (Survey 1 or Survey 2). Quantile regression models can be found in S3 Text.

[b] The corrected median difference as percentage of the maximum score that can be obtained on the scale.

[c] The p-value was calculated using quantile regression in which we corrected for differences in demographic information between the participants in Survey 1 and Survey 2. NS = not significant (p>0.05).

30 that can be obtained) and -2.0 for the SDS (a change of 10% of the total score of 21 that can be obtained). The largest corrected median difference on the KAP was found for persons affected by leprosy (corrected median difference 3.0, 38% of total score, p<0.001). With corrected median differences of -10.0 and -10.3 (33–34% of the total score), the corrected median differences for the EMIC-CSS were larger in Chandauli district than in Fatehpur district. The largest corrected median differences (-3.0 and -2.9 or 14% of the total score of 21) for the SDS were found among contacts and community members in Chandauli and among health workers in Fatehpur (Table 3).

Multivariate analysis showed that the determinants of leprosy knowledge and community stigma were comparable for Survey 1 and Survey 2. In Survey 2, income, caste and 'having seen posters' were also included in the models. 'Having seen posters' alone had a larger effect on the KAP scores (R-squared = 0.15, univariate analysis) than on the EMIC-CSS and SDS scores (R-squared = 0.05 and 0.06 respectively, univariate analysis). The models for Survey 2 explained more of the variability of knowledge and stigma (Table 5). An overview of the full multivariate regression models can be found as supporting information file S4 Text.

In the in-depth interviews and focus group discussions, many participants indicated that there had been a change in perception in the community. Some participants indicated that people in the community used to believe something, but not anymore. Some participants related this change in perception of the community to (knowing about) preventive medication. One close contact explained:

*". . .Earlier people used to behave [negative] like this, now people have started understanding that untouchability does not happen [referring to transmission]. . ."* (30-year old close contact, male, FGD, Chandauli)

**Table 4. Corrected median differences in EMIC-CSS (range 0–30) and SDS (range 0–21) scores between Survey 1 (n = 867) and Survey 2 (n = 641).**

| Dataset | EMIC-CSS | | | | | SDS | | | | |
|---|---|---|---|---|---|---|---|---|---|---|
| | Survey 1, median (Q1-Q3) | Survey 2, median (Q1-Q3) | Corrected median difference[a] | Change in score[b] | p-value[c] | Survey 1, median (Q1-Q3) | Survey 2, median (Q1-Q3) | Corrected median difference[a] | Change in score[b] | p-value[c] |
| *Whole dataset (n = 1067~842)* | **17.0 (11.0–21.0)** | **9.0 (5.0–14.0)** | **-7.3** | 24.3% | **<0.001** | **5.0 (3.0–10.0)** | **3.0 (1.0–6.0)** | **-2.0** | 9.5% | **<0.001** |
| Contacts and community (n = 767~541) | 17.0 (11.0–21.0) | 9.0 (6.0–14.0) | -7.3 | 24.3% | <0.001 | 6.0 (3.0–10.0) | 4.0 (2.0–7.0) | -2.7 | 12.9% | <0.001 |
| Health workers (n = 100~100) | 15.0 (10.0–21.8) | 6.0 (3.0–13.0) | -4.3 | 14.3% | <0.001 | 2.0 (0.0–6.0) | 0.0 (0.0–2.0) | -1.0 | 4.8% | 0.044 |
| *Chandauli district (n = 621~420)* | **18.0 (11.0–22.0)** | **7.0 (5.0–9.0)** | **-10.0** | 33.3% | **<0.001** | **5.0 (2.0–9.0)** | **3.0 (1.0–4.0)** | **-2.0** | 9.5% | **<0.001** |
| Contacts and community (n = 471~270) | 18.0 (12.0–22.0) | 8.0 (5.8–10.0) | -10.0 | 33.3% | <0.001 | 5.0 (3.0–10.0) | 3.0 (2.0–5.0) | -3.0 | 14.3% | <0.001 |
| Health workers (n = 50~50) | 15.0 (8.8–23.3) | 4.0 (2.0–6.0) | -10.3 | 34.3% | <0.001 | 1.0 (0.0–3.3) | 0.0 (0.0–1.0) | 0.0 | 0.0% | NS |
| *Fatehpur district (n = 446~422)* | **15.0 (11.0–20.0)** | **14.0 (7.0–19.0)** | **-1.7** | 5.7% | **0.029** | **6.0 (3.0–10.0)** | **4.0 (0.0–9.0)** | **-2.3** | 11.0% | **<0.001** |
| Contacts and community (n = 296~271) | 15.0 (10.3–21.0) | 14.0 (7.0–19.0) | -2.0 | 6.7% | 0.018 | 6.0 (3.0–11.0) | 5.0 (1.0–10.0) | -2.3 | 11.0% | <0.001 |
| Health workers (n = 50~50) | 14.5 (11.0–20.0) | 13.0 (6.0–18.5) | -1.6 | 5.3% | NS | 3.5 (1.0–6.0) | 0.0 (0.0–3.0) | -2.9 | 13.8% | <0.001 |

[a] We corrected (adjusted) for age, sex, district, education, religion, participant type and data collection period (Survey 1 or Survey 2). Quantile regression models can be found in S3 Text.

[b] The corrected median difference as percentage of the maximum score that can be obtained on the scale.

[c] The p-value was calculated using quantile regression in which we corrected for differences in demographic information between the participants in Survey 1 and Survey 2. NS = not significant (p>0.05).

Furthermore, the transcripts of the interviews and focus group discussions revealed that over half of the participants knew that leprosy is caused by bacteria, and almost all participants mentioned loss of sensation and/or skin patches as early symptoms of leprosy and knew leprosy can be treated with medication. Knowledge about treatment, cause and symptoms was good. There were still some misconceptions regarding the cause of leprosy. For example, some participants thought leprosy is caused by a blood or vitamin deficiency, dirt or being unclean.

Approximately half of the participants indicated that (some) community members discriminate or keep a distance from persons affected by leprosy. Most participants said these community members behave this way because they have incorrect or insufficient knowledge about leprosy or because they are afraid of getting infected by the disease themselves. One participant explained:

> ". . .Most people discriminate because they do not know about this disease, they feel that it is an untouchable disease, whereas this is not true. . .." (58-year old close contact, male, IDI, Chandauli)

Some participants stressed that community members only discriminate if leprosy is visible or if persons affected by leprosy are not treated. Approximately a quarter of all participants said that there is no discrimination, that community members behave well or normal towards

**Table 5. Correlations between level of knowledge (KAP score) about leprosy, community stigma (EMIC-CSS), social distance (SDS) and the other variables in the dataset including data of persons affected by leprosy, close contacts and community members.** Participant type and district were included in all models to control for confounding.

| Questionnaire | Survey 1* | | Survey 2** | |
|---|---|---|---|---|
| | Variables included in the model | R-squared | Variables included in the model | R-squared |
| KAP measure (knowledge of leprosy) | (Participant type, district)<br>No (formal) education<br>Higher education | 0.054 | (Participant type, district)<br>No (formal) education<br>Higher education<br>Income less than 5,000<br>Has seen PEP++ posters | 0.355 |
| EMIC-CSS (community stigma) | (Participant type, district) | 0.105 | (Participant type, district)<br>Primary education<br>Knowledge about leprosy (KAP)<br>Income less than 1,000<br>Has seen PEP++ posters | 0.292 |
| SDS (social distance as a proxy for attitudes) | (Participant type, district)<br>No (formal) education<br>Gender<br>Knowledge about leprosy (KAP) | 0.050 | (Participant type, district)<br>No (formal) education<br>Primary education<br>Knowledge about leprosy (KAP)<br>Occupation paid work<br>Income 5,001 to 10,000<br>Has seen PEP++ posters | 0.232 |

* Variables included: participant type, district, age, gender, education, occupation, and for the EMIC-CSS and SDS also 'KAP score' and 'knowing someone affected by leprosy'.

** Variables included: participant type, district, age, gender, education, occupation, marital status, monthly household income, caste, having seen PEP++ posters, and for the EMIC-CSS and SDS also 'KAP score' and 'knowing someone affected by leprosy'.

persons affected by leprosy. Many of the participants who mentioned there is no discrimination, also mentioned that they advise persons affected by leprosy to get treatment. A few participants explicitly stated that they don't think persons affected should be discriminated.

> "...They [community] behave differently, like talking with them [persons affected by leprosy] by keeping a distance etc. According to me, this is wrong, there should be no discrimination against them..." (51-year-old community member, female, IDI, Fatehpur)

Another participant explained:

> "...People of the community do not discriminate. Everyone sits together and tells [the person affected by leprosy] to get treatment for leprosy..." (32-year-old close contact, male, IDI, Chandauli)

Finally, almost all participants had heard about post-exposure prophylaxis (PEP) and the PEP++ project. Everyone who knew about PEP had a positive attitude towards it. Many participants indicated that knowing about PEP and the possibility of PEP positively changed the perception of leprosy. A person affected by leprosy explained:

> "...There is a change in thinking that now if you take [preventive] medicine before [you have symptoms] then there will be no disease. The medicine that prevents leprosy is a very good idea..." (19-year-old person affected by leprosy, female, IDI, Chandauli)

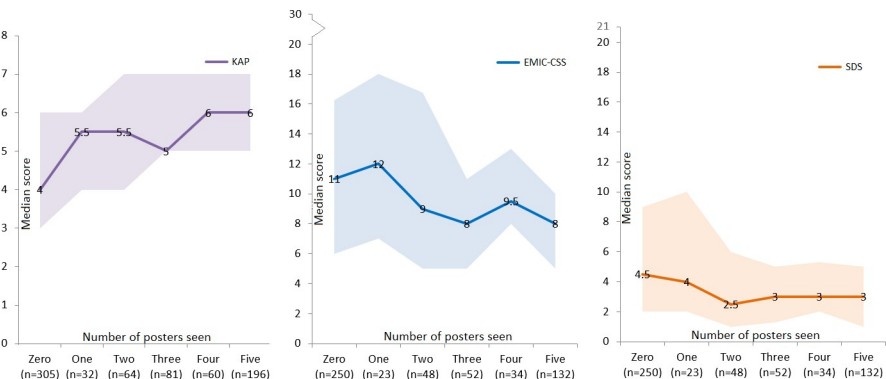

**Fig 3. An overview of the number of posters seen and the corresponding median KAP (knowledge, ranging from 0–8 with higher scores denoting better knowledge), EMIC-CSS (community attitudes, ranging from 0–30 with higher scores denoting more negative attitudes) and SDS scores (personal attitudes, ranging from 0–21 with higher scores denoting more negative attitudes) and corresponding first and third quartiles.** Please note that the graphs have different y-axes.

## Impact of the posters

Most participants (health care workers excluded) indicated they had seen a poster in the villages (34%, n = 287) at the health facility (26%, n = 220) or in public transport (10%, n = 84). Almost two-third of the participants were able to identify at least one poster (61%, n = 482). Participants correctly identified two posters on average. A total of 305 participants (36.2%) had not seen any poster and 196 participants (23.3%) had seen all five posters.

Persons affected by leprosy, close contacts and community members were also shown five posters and asked if they had seen them. Between 38% (n = 281, fifth poster) and 48% (n = 353, first poster) of the participants indicated that they had seen one of the five posters. Participants in Fatehpur (n = 369) were also shown a poster that was never used and asked whether they had seen it. Two participants (0.5%) thought they had seen this poster and 367 participants (99.5%) said they had not seen the poster. Fig 3 gives an overview of the number of posters the participants had seen and their mean KAP, EMIC-CSS and SDS scores. There was an association between the number of posters seen and the KAP (n = 738, rho = 0.389, p<0.001), EMIC-CSS (n = 539, rho = -0.208, p<0.001) and SDS (n = 539, rho = -0.203, p<0.001) scores in Survey 2.

## Discussion

### Salient results

Findings from this study suggest that the contextualized posters and community meetings were effective in increasing knowledge of leprosy (a change of 19% of the maximum KAP score), in changing community attitudes (a change of 24% on the maximum EMIC-CSS score), and in changing personal attitudes (a change of 10% on the maximum SDS score) of all participant groups. In addition, when we used a cut-off point to determine adequate knowledge of leprosy and stigmatizing attitudes towards leprosy, the percentage of participants with adequate knowledge of leprosy was 13% in Survey 1 and 53% in Survey 2, and the percentage of participants with stigmatizing attitudes was 86% in Survey 1 and 61% in Survey 2, and the percentage of participants with negative personal attitudes was 37% in Survey 1 and 19% in Survey 2. We consider the effect high for knowledge of leprosy and community attitudes, and moderate for personal attitudes. It is likely that the change in community attitudes was greater

than in personal attitudes, because the social distance score (SDS, personal attitudes) was already relatively low in Survey 1, so could decrease less.

The largest effect of the whole package of interventions was on knowledge of leprosy of persons affected by leprosy (a change of 38% of the maximum score) and on community attitudes of contacts and community members (a change of 24% of the maximum score). From other studies, we know that it is easier to improve knowledge than to change behavior [30, 31], our findings are therefore very encouraging. It is possible that the effect of our interventions is an underestimation, because the post-intervention participants had lower education levels than the pre-intervention participants. This could have influenced our results, since less education was associated with less knowledge about leprosy and more negative attitudes.

The smallest effect of the whole package of interventions was seen among knowledge and personal attitudes of health workers. It is likely that the effect on health workers was smaller because they were not specifically targeted with the posters and community meetings. In addition, before the implementation of the interventions their knowledge of leprosy was already better and their attitudes more positive compared to the other participants in our study. Their scores could therefore increase (knowledge) or decrease (stigma) less. Health workers were asked to support and be involved in preparatory activities for the PEP++ project, such as finding houses of persons affected by leprosy and giving feedback on the posters, and they were informed about the project and preventive medication. In addition, they were exposed to the posters at the health centers daily. We assume that this involvement and exposure has positively influenced their perceptions.

Surprisingly, the effect of the total package of interventions on community stigma was larger in Chandauli district than in Fatehpur district. It is possible that the difference can partly be explained by the religious meetings featuring local artists (Shiv charcha) that were held only in Chandauli district in addition to the community meetings. In addition, according to the district teams, media exposure of the project and government interest and commitment were greater in Chandauli district. In Chandauli district, 52 news articles covered announcements of the community meetings and a brief explanation of the project, while this was covered in 26 news articles in Fatehpur district. The way persons affected are portrayed in the media reflects, defines, and perpetuates public perceptions of those who are portrayed [32]. In other fields, while media have been found to be a source of stigma through the negative portrayal of persons affected, they have also been found to reduce stigma by raising awareness (as was done in our study) [33–35]. However, we cannot offer a definitive explanation of the difference the interventions had on community stigma between the two districts.

From the exploratory and other studies, we know that knowledge about leprosy plays a crucial role in stigma [8, 36–39]. While knowledge gaps can be addressed by information, to change attitudes and perceptions is more difficult and requires a combination of health education and behavioral change interventions [40, 41]. Good knowledge of leprosy does not necessarily lead to more positive attitudes toward persons affected by leprosy [12]. Interestingly, the determinants of leprosy knowledge (education, income, exposure to posters) and community stigma (education, leprosy knowledge, income, occupation, exposure to posters) in the present study were similar before and after the interventions had been implemented. We collected additional information in the post-intervention measurement: income, caste and 'having seen posters' and included this in our analysis. The post-intervention models explained more of the variability of knowledge and stigma. This is in part explained by the effect of income and exposure to the posters.

In the present study there was a positive association between the number of posters seen and the level of knowledge and positive attitudes towards persons affected by leprosy. The more posters participants indicated to have seen, the better their knowledge and the more

positive their attitudes. However, since we didn't assess the knowledge and attitudes of the participants in the second survey before exposure to the posters, it is uncertain that the impact can be attributed to the posters. Nevertheless, the correlation with the number of posters seen is strongly suggestive of such an effect. Beliefs about leprosy are often deeply rooted in people's culture [12]. To address this, we focused on local beliefs and misconceptions and have consulted the target populations in selecting the interventions and developing the posters. Printed media like posters, billboard and leaflets have been used to increase community awareness and reduce leprosy-related stigma in other studies also, but their impact has not been evaluated rigorously [16, 42–44]. Although written materials, like posters and leaflets, are not the most suitable approach for populations with low educational levels [45], the contextualization and careful pretesting of the imagery used appears to have resulted in a positive impact on community knowledge and perception in the study.

## Methodological considerations

A key feature of the interventions in the present study is that they are contextualized, relatively low-cost and easy to replicate. We ensured that the materials and messages were targeted and contextualized and that relevant topics were prioritized, e.g., cause, mode of transmission, symptoms and infectiousness of leprosy. Contextualized materials and messages are more effective than generic messages [46]. In addition, community consultation and involvement was used—this is more episodic community participation, in contrast to, for example, community engagement, which suggests an ongoing and active relationship [11]. The interventions were developed through collaborations and consultations between the target population, including persons affected by leprosy, researchers, health workers, leprosy experts, communication experts and policymakers. This maximizes the likelihood that the interventions fit with the target groups' needs and acceptability, and the uptake of the interventions by policymakers [23].

Several successful community-based stigma reduction interventions have been conducted in the field of leprosy, all including elements of community participation or engagement, such as informing, consulting, involving, collaborating and empowering communities [10, 11]. These studies have shown encouraging results. Successful community-based stigma reduction interventions in the field of leprosy include education and counselling through (community) stigma reduction committees [9], stigma reduction interventions through groups of health workers, volunteers and persons affected in self-help groups [8], and rights-based counselling and contact events [4]. Indeed, community participation and engagement can ensure that research is relevant and impactful. Community engagement has been successful for control and elimination of other diseases also, such as malaria [12, 13]. In the present study, in addition to the community's involvement in the development of the interventions, the community was also consulted and discussions were held in community meetings for the purpose of education and changing negative attitudes regarding leprosy. Efforts were made to ensure community and health worker engagement in the interventions, by consultation and by involving them in preparatory meeting.

We were not able to determine the actual change in behavior, given that there are no suitable measures to assess this. However, we measured knowledge, attitude and perceived practices and these measures in part reflect actual behavior (e.g., a person's perception of their behavior in a given situation), and our qualitative data show reflections of participants, indicating an actual change in perception and behavior. We therefore conclude that it is likely that in addition to a change in perceptions, there was likely also a change in behavior after the interventions. We would recommend that future studies explore meaningful ways to assess actual

changes in behavior and indicators of behavior change, for example by asking persons affected by leprosy about their experiences at health facilities.

A novel feature of our study is that we determined and used a cut-off point for negative personal attitudes on the SDS, which was not yet available. A cut-off point for positive/negative attitudes is important, because it helps readers and practitioners to interpret the findings. It can also be used to estimate the magnitude of (meaningful) effect.

This study has several limitations. First, a randomized controlled design was not feasible given the nature of our intervention, namely, community-wide poster dissemination and meetings. Instead, we used a pre/post intervention design. It is possible that not all of the observed changes are due to the interventions–some changes may have been caused by other factors. We tried to minimize this by selecting a random sample. However, we cannot rule out other factors that may have contributed to the outcomes observed. While it is unclear how much of the effect found can be attributed to the interventions, given our findings it is very likely that the interventions have contributed to the outcomes. Second, there were differences between sociodemographic characteristics of the pre- and post-intervention participant groups. This was especially the case for close contacts and community members, with post-intervention participants having lower education levels. We have corrected for these differences in our analysis, but because of this, we were unable to use a standardized measure of effect size. This made it more difficult to determine the magnitude of the effect of the interventions. We recommend separately evaluating each element of an intervention in future studies (instead of the whole package of interventions), to gain a better understanding of the impact of each element. Finally, it would have been interesting if we had evaluated the impact of the community meetings on knowledge and perceptions of influential people specifically, they were not included as a separate target group in the surveys (evaluation) of the interventions.

## Conclusions

The contextualized posters and community meetings in this study were effective in increasing leprosy-related knowledge and changing perceptions of leprosy in Fatehpur and Chandauli districts in Uttar Pradesh, India. The interventions in this study are relatively low-cost and are easy to replicate. Given that changing attitudes and perceptions is difficult and generally requires a combination of health education and behavioral change interventions, the results are very encouraging. Future studies should explore meaningful ways to assess actual changes in behavior and indicators of behavior change. In addition, the long-term effect of the interventions should be studied.

## Supporting information

**S1 Text. Supporting information file–intervention development based on 6SQuID.**
(DOCX)

**S2 Text. Posters and English translation of the posters.**
(DOCX)

**S3 Text. Quantile regression models.**
(DOCX)

**S4 Text. Supporting information file–multivariate regression models KAP, EMIC-CSS and SDS.**
(DOCX)

## Acknowledgments

We are grateful to the contributions of all of the participants. We thank the research assistants who collected the data for this study and who implemented the interventions. We thank data entry operators for entering the data. We want to thank the members of the IEC working group, Dr. P.R. Manglani, Mr. Jacob Oommen and Ms. Soumya Jha for their valuable input during the development of the interventions. We gratefully acknowledge the support of Prof. Jugal Kishore, the Principal Investigator of the PEP++ project. Finally, we are grateful to Dr. Daan Nieboer for his statistical advice and support.

## Author Contributions

**Conceptualization:** Anna T. van 't Noordende, Suchitra Lisam, Duane C. Hinders, Wim H. van Brakel.

**Data curation:** Anna T. van 't Noordende, Vivek Singh, Atif Sadiq.

**Formal analysis:** Anna T. van 't Noordende.

**Funding acquisition:** Jan Hendrik Richardus, Wim H. van Brakel.

**Investigation:** Anna T. van 't Noordende, Suchitra Lisam, Vivek Singh, Atif Sadiq, Duane C. Hinders, Wim H. van Brakel.

**Methodology:** Anna T. van 't Noordende, Duane C. Hinders, Jan Hendrik Richardus, Wim H. van Brakel, Ida J. Korfage.

**Project administration:** Suchitra Lisam, Vivek Singh, Atif Sadiq, Ashok Agarwal, Duane C. Hinders.

**Resources:** Suchitra Lisam, Vivek Singh, Atif Sadiq, Ashok Agarwal, Duane C. Hinders, Wim H. van Brakel.

**Software:** Anna T. van 't Noordende.

**Supervision:** Suchitra Lisam, Vivek Singh, Atif Sadiq, Ashok Agarwal, Duane C. Hinders, Jan Hendrik Richardus, Wim H. van Brakel, Ida J. Korfage.

**Validation:** Anna T. van 't Noordende, Suchitra Lisam, Vivek Singh, Atif Sadiq, Ashok Agarwal, Duane C. Hinders, Jan Hendrik Richardus, Wim H. van Brakel, Ida J. Korfage.

**Visualization:** Anna T. van 't Noordende.

**Writing – original draft:** Anna T. van 't Noordende.

**Writing – review & editing:** Anna T. van 't Noordende, Suchitra Lisam, Vivek Singh, Atif Sadiq, Ashok Agarwal, Duane C. Hinders, Jan Hendrik Richardus, Wim H. van Brakel, Ida J. Korfage.

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
