## [Decision Letter · Decision Letter 0]

23 May 2021

Dear Ms. van 't Noordende,

Thank you very much for submitting your manuscript "Changing perception and improving knowledge of leprosy: an intervention study in Uttar Pradesh, India" for consideration at PLOS Neglected Tropical Diseases. As with all papers reviewed by the journal, your manuscript was reviewed by members of the editorial board and by several independent reviewers. In light of the reviews (below this email), we would like to invite the resubmission of a significantly-revised version that takes into account the reviewers' comments. 

We cannot make any decision about publication until we have seen the revised manuscript and your response to the reviewers' comments. Your revised manuscript is also likely to be sent to reviewers for further evaluation.

Sincerely,

David John Chandler, MB ChB, DTM&H, MSc, MRCP

Guest Editor

Michael Marks

Deputy Editor

Reviewer's Responses to Questions

**Key Review Criteria Required for Acceptance?**

**Methods**

-Are the objectives of the study clearly articulated with a clear testable hypothesis stated?

-Is the study design appropriate to address the stated objectives?

-Is the population clearly described and appropriate for the hypothesis being tested?

-Is the sample size sufficient to ensure adequate power to address the hypothesis being tested?

-Were correct statistical analysis used to support conclusions?

-Are there concerns about ethical or regulatory requirements being met?

Reviewer #1: Objective of the study is not clear. concepts in the title and used in the manuscript -perception, knowledge, attitudes and practices.

The design is not clear, as presented not appropriate to test the change.

Sampling procedure is not explained. Two districts Chandauli (population 1.95 million) and Fatehpur (population 2.63 million)are studied. How arrived at total of 1067 participants in the pre and 843 at post intervention surveys is to be clearly explain

The statistical analysis does not measure the change due to the intervention because no control area is mention in the desion.

Reviewer #2: Clear objectives

Appropriate study design for the objectives

study population and sampling described in previous publication

ethics = satisfactory

Reviewer #3: (No Response)

**Results**

-Does the analysis presented match the analysis plan?

-Are the results clearly and completely presented?

-Are the figures (Tables, Images) of sufficient quality for clarity?

Reviewer #1: No analysis plan is mentioned in the methodology

Results presented are of only intervention area.

Reviewer #2: Comprehensive analysis

clearly presented results

Good quality figures

Errors found:

line 280 typographical error 'martial' should be 'marital'

line 325 table 5

line 240 = table 5 (duplicate number for two different tables

Reviewer #3: (No Response)

**Conclusions**

-Are the conclusions supported by the data presented?

-Are the limitations of analysis clearly described?

-Do the authors discuss how these data can be helpful to advance our understanding of the topic under study?

-Is public health relevance addressed?

Reviewer #1: Data presented do not support the conclusions.

Reviewer #2: Conclusion supported by data presented

Limitations explained clearly

Reviewer #3: (No Response)

**Editorial and Data Presentation Modifications?**

Reviewer #1: The results can pre presented as a survey findings, cant be presented as a change of an intervention as no control area data is presented

Reviewer #2: minor edits as mentioned above - ACCEPT

Reviewer #3: (No Response)

**Summary and General Comments**

Reviewer #1: Need to be modified drastically as per the new manuscript

Reviewer #2: (No Response)

Reviewer #3: Noordende and colleagues have reported an important study around intervention that contributed in improving knowledge of leprosy in Uttar Pradesh, India. The study is important and adds to the current evidence. I have comments and suggestions to improve the content and presentations as below. 

General

• Without a schematic diagram of how study spanned over in time and the intervention, it is difficult for readers to visualize how they study might have taken place. I urge authors to present the study activities using a schematic diagram that can describe interventions, and evaluation in time period. You can change Table 2 to a figure.

• I think authors are describing great interventions and their impacts, but the narrow scope of discussion may limit their readability. I urge authors to expand on it more, with more literature and disciplinary terminologies, so that its implications are transferrable to other diseases and conditions.

Specific

• Regarding impact of interventions: posters and community meetings. Authors have tried to evaluate the impact of these interventions with the outcome measures as increase in knowledge and reduction in stigmatizing attitudes. First of all, somewhere it needs to be clearly acknowledged, with such interventions, and owing to the fact that they are based on pre- and post-intervention, the chances that we may over-claim the effects/impacts are high. I would generally acknowledge them by stating clearly in the limitation section about attribution versus contribution. Clearly these interventions may have contributed, but we cannot rule out other myriad factors that may have contributed to achieve the outcome. 

• While authors have very well described the interventions: posters, and community meetings. These activities, by virtue of their characteristics are included under the terminology of ‘community engagement’. Indeed, authors have clearly generated evidence around how contact intervention in the past have affected the stigma reduction, that may well be described as one of the strand or aspect of community engagement. It is essential to appreciate the term ‘community engagement’ for several reasons including, how we may have been discussing the same content but through different lenses. This also means, interventions such as ‘posters’ and ‘community meetings’ are discussed and evaluated as essential elements of community engagement in various diseases, such as malaria. Certainly, discussing these relevant literatures from other diseases will expand the scope of the current paper. This may further make other disciplinary researchers to learn from your evaluation as well. Eventually, transferability of knowledge generated in one particular disease may well be an important element for researchers working in other diseases. 

• I recommend authors to explain more in detail how the community meetings were held, what was the characteristic features of these meetings? Were they dialogues, monologues, presentation? Please explain the content in more details. Same for the posters, I would recommend authors to present the posters in the main manuscript, so that readers get to know what was the interventions rather than to speculate about it. 

• Authors have stated throughout the manuscript, that contextualized messages (posters and meetings) are important. I cannot agree more, I applaud for their statement. I urge authors to illustrate such contextualization more explicitly, with more examples if possible?

• There are recent suggestions/recommendations regarding the need for increase community engagement for stigma reduction strategies among community members and health care providers, authors need to acknowledge these literatures, challenge or discuss the possibilities.

PLOS authors have the option to publish the peer review history of their article (what does this mean?). If published, this will include your full peer review and any attached files.

Reviewer #1: Yes: Raju MS

Reviewer #2: No

Reviewer #3: No
---

## [Decision Letter · Decision Letter 1]

16 Jul 2021

Dear Ms. van 't Noordende,

We are pleased to inform you that your manuscript 'Changing perception and improving knowledge of leprosy: an intervention study in Uttar Pradesh, India' has been provisionally accepted for publication in PLOS Neglected Tropical Diseases.

Best regards,

David John Chandler, MB ChB, DTM&H, MSc, MRCP

Guest Editor

Michael Marks

Deputy Editor

Reviewer's Responses to Questions

**Key Review Criteria Required for Acceptance?**

**Methods**

-Are the objectives of the study clearly articulated with a clear testable hypothesis stated?

-Is the study design appropriate to address the stated objectives?

-Is the population clearly described and appropriate for the hypothesis being tested?

-Is the sample size sufficient to ensure adequate power to address the hypothesis being tested?

-Were correct statistical analysis used to support conclusions?

-Are there concerns about ethical or regulatory requirements being met?

Reviewer #2: (No Response)

Reviewer #3: Yes

**Results**

-Does the analysis presented match the analysis plan?

-Are the results clearly and completely presented?

-Are the figures (Tables, Images) of sufficient quality for clarity?

Reviewer #2: (No Response)

Reviewer #3: Yes

**Conclusions**

-Are the conclusions supported by the data presented?

-Are the limitations of analysis clearly described?

-Do the authors discuss how these data can be helpful to advance our understanding of the topic under study?

-Is public health relevance addressed?

Reviewer #2: (No Response)

Reviewer #3: Yes

**Editorial and Data Presentation Modifications?**

Reviewer #2: (No Response)

Reviewer #3: (No Response)

**Summary and General Comments**

Reviewer #2: (No Response)

Reviewer #3: Authors have diligently revised the manuscript.

PLOS authors have the option to publish the peer review history of their article (what does this mean?). If published, this will include your full peer review and any attached files.

Reviewer #1: No

Reviewer #2: No

Reviewer #3: No

---

## [Editor Report · Acceptance letter]

19 Aug 2021

Dear Ms. van 't Noordende,

We are delighted to inform you that your manuscript, "Changing perception and improving knowledge of leprosy: an intervention study in Uttar Pradesh, India," has been formally accepted for publication in PLOS Neglected Tropical Diseases.

Best regards,

Shaden Kamhawi

co-Editor-in-Chief

Paul Brindley

co-Editor-in-Chief
